# Enablers and Barriers of COVID-19 Vaccination in the Philippines

**DOI:** 10.3390/vaccines13070719

**Published:** 2025-07-01

**Authors:** Evalyn Roxas, Paulyn Jean Acacio-Claro, Maria Margarita Lota, Alvin Abeleda, Soledad Natalia Dalisay, Madilene Landicho, Yoshiki Fujimori, Jan Zarlyn Rosuello, Jessica Kaufman, Margaret Danchin, Vicente Belizario, Florian Vogt

**Affiliations:** 1College of Public Health, University of the Philippines Manila, Manila 1000, Philippines; paclaro@up.edu.ph (P.J.A.-C.); mmlota@up.edu.ph (M.M.L.); aiabeleda@up.edu.ph (A.A.); yrfujimori@up.edu.ph (Y.F.); jarosuello@up.edu.ph (J.Z.R.); vybelizario@up.edu.ph (V.B.J.); 2Unit of Health Sciences, Faculty of Social Sciences, Tampere University, 33520 Tampere, Finland; 3Department of Anthropology, College of Social Sciences and Philosophy, University of the Philippines Diliman, Quezon City 1101, Philippines; smdalisay@up.edu.ph (S.N.D.); mblandicho@up.edu.ph (M.L.); 4Third World Studies Center, College of Social Sciences and Philosophy, University of the Philippines Diliman, Quezon City 1101, Philippines; 5Murdoch Children’s Research Institute, Royal Children’s Hospital, Parkville, VIC 3052, Australia; jess.kaufman@mcri.edu.au (J.K.); margie.danchin@rch.org.au (M.D.); 6Department of Paediatrics, University of Melbourne, Melbourne, VIC 3010, Australia; 7The Kirby Institute, University of New South Wales, Sydney, NSW 2033, Australia; fvogt@kirby.unsw.edu.au; 8National Centre for Epidemiology and Population Health, Australian National University, Canberra, ACT 2601, Australia

**Keywords:** COVID-19, vaccination, vaccine uptake, Philippines

## Abstract

**Background**: The COVID-19 pandemic prompted extensive vaccination efforts globally, yet in the Philippines, many families remained unvaccinated. Caregivers are key decision-makers for family vaccination, but evidence on factors influencing their own vaccine uptake is limited. **Methods**: A cross-sectional survey of primary caregivers was conducted in low COVID-19 vaccine uptake regions in the Philippines from July to October 2023 using a validated questionnaire. Multivariable logistic regression identified enablers and barriers to vaccine uptake. **Results**: Among 775 respondents, 72.3% completed primary vaccination, 3.3% had incomplete vaccination, and 24.4% were unvaccinated. Key factors for vaccination included self, family, and community protection, and the influence of government regulations. Distrust in vaccine safety was the main barrier. Positive associations with vaccine uptake were found for age [30–45 years (aOR = 2.23) and 46–59 years (aOR = 2.84)], education [secondary (aOR = 2.25) and tertiary (aOR = 4.93)], and employment (aOR = 1.99). Confidence in vaccine safety (aOR = 1.92), vaccine effectiveness (aOR = 2.23), and satisfaction with vaccination efforts (aOR = 2.39) were additional enablers. Disagreement with restrictions on the unvaccinated was a barrier (aOR = 0.31). **Conclusions**: This study identified multiple factors influencing COVID-19 vaccination among primary caregivers in low uptake areas of the Philippines. Interventions addressing perceptions about vaccine safety and effectiveness, particularly among younger and less educated caregivers, may improve public trust and satisfaction with vaccination efforts.

## 1. Introduction

The COVID-19 pandemic posed unprecedented challenges in the health system globally. Over 4 million cases of COVID-19 were recorded in the Philippines from the beginning of the pandemic in 2020 to the end of 2023. Most cases recorded were in the 20–29 and 30–39 age groups [1]. However, the greatest number of mortalities recorded were in the 65 and above age group. The rapid development of safe and effective vaccines became one of the most important interventions. By 2023, two-thirds (67%) of the global population was vaccinated with a complete primary series of COVID-19 vaccines, but only a third (32%) had received at least one booster dose [2]. In the Philippines, the COVID-19 vaccination was rolled out through a multi-sectoral approach involving different departments, agencies, and non-governmental organizations, experts, and professionals in an effort to provide free, safe, and effective vaccines [3]. The initial or phase 1 roll-out of the vaccine in March 2021 prioritized the population with the highest risk such as the frontline healthcare workers, senior citizens, and adults with comorbidities [4]. Subsequently, the phase 2 roll-out for other essential workers, followed by the phase 3 roll-out for the rest of the population was conducted. Towards the end of 2023, the country reached the WHO target of 70% coverage for the primary series [2]. Vaccination efforts in the Philippines have led to a decrease in the number of new cases per day leading to the declaration of the end of the COVID-19 public health emergency in July 2023 [5]. Despite this achievement, the Philippines only ranked eighth out of the eleven member countries of the Association of South East Asian Nations (ASEAN) in COVID-19 vaccine coverage for the primary series, and last for the booster series [6].

Certain sociodemographic characteristics like age, educational attainment, employment status, and household economic status have been suggested as determinants of COVID-19 vaccine uptake, with substantial heterogeneity and inconsistencies [7,8,9,10]. Aside from these, several other factors influence the uptake of COVID-19 vaccines, such as perceptions on disease risk, perceptions on vaccine safety and effectiveness, the direct and indirect costs of vaccination, and levels of trust in government authorities and healthcare workers [11]. There is a need to better understand the combination of these factors specifically in the Philippines, where challenges in vaccine uptake and confidence have long historical precedence [12].

In the Philippines, primary caregivers of children are often the key decision-makers for the vaccination of the entire household. Most caregivers consider vaccination an integral part of their responsibility for the health of their household members [13]. A better understanding of enablers and barriers to COVID-19 vaccination among primary caregivers will therefore provide much-needed evidence for better targeted public health interventions to increase the uptake of COVID-19 vaccines and other vaccines, in the Philippines. Therefore, in this study, we aimed to identify factors that affect COVID-19 vaccine uptake among primary caregivers in regions with low vaccine uptake.

## 2. Materials and Methods

A cross-sectional community-level paper-based survey was conducted among primary caregivers from July to October 2023 to determine enablers and barriers of COVID-19 vaccination. In consultation with the Department of Health (DOH)—Public Health Operations Center (PHOC), three regions, namely Regions IV-B (MIMAROPA), V (Bicol), and VIII (Eastern Visayas), were selected as study sites due to lower COVID-19 vaccine coverage as compared to other regions. The selection of these regions was based on the vaccine uptake in August 2022, at which point the study sites had not yet met the target vaccination rate of 70% [2] and had approximately 10–12% booster dose coverage. Employing a multi-stage stratified cluster sampling, regions were further stratified into provinces, municipalities, and barangays (communities). In coordination with the DOH Centers for Health Development (CHD) of each region, two provinces and two municipalities from each province were selected based on low COVID-19 vaccine coverage. Lastly, the Rural Health Units (RHUs) of the identified municipalities selected two barangays as study sites. In total, there were 24 barangays included as study sites. Based on a coverage rate of 10.3%, a design effect of 2, a desired absolute precision of 6%, and adding a 30% buffer, the total study sample size calculated was 768.

Before field visits for data collection, community health workers went from house-to-house in the barangays to recruit primary caregivers residing in the community aged 18 years and above with at least one child aged 0–11 years to take part in the study. The recruited participants were also screened by the research team to ensure that the inclusion criteria were met. Those aged less than 18 years and that were not primary caregivers of children aged 0–11 at the time of the study were excluded. All eligible caregivers present in the venue (e.g., health facility or multipurpose hall) during the actual data collection were included in the study until the quota per barangay needed to meet the sample size per province was attained.

The survey questionnaire was developed in reference to the Centers for Disease Control Vaccine Confidence Survey Question Bank [14] and the WHO-Strategic Advisory Group of Experts (SAGE) Vaccine Hesitancy Matrix Questions [15] and underwent two rounds of face validity testing. Variables collected were sociodemographic characteristics of adults such as age in years, sex (male or female), civil status (single, married, live-in, separated, or widowed), religion (Catholic, other Christian denomination, Islam, and others), educational attainment (elementary, high school, or college level) and employment status (unemployed, student, pensioner, self-employed, or employed). Municipal class as determined by the average annual income of the municipality of the respondents was also collected [16]. Reasons for receiving or declining the vaccine, perceptions on COVID-19 infection and vaccination, and sources of information on vaccination were also collected. Self-reported uptake of COVID-19 vaccines (unvaccinated/incomplete primary series and complete primary series with or without booster) was measured as an outcome variable. Other personally identifiable information (e.g., name, address, and contact details) were not collected to ensure the anonymity of the survey participants.

Two separate logistic regression models, one containing sociodemographic variables, and another containing all statements on perceptions about COVID-19 infection and vaccination were first tested. All variables including missing information on perceptions about COVID-19 infection and vaccination were included in the analysis. Using a backward elimination approach, statistically significant variables (*p* < 0.05) in each model were retained and combined in a third model for analyses. The final logistic regression model consisted of either variables that were statistically significant (*p* < 0.05) or were considered essential based on likelihood ratio test (LRT) results (*p* < 0.05). This model was adjusted to account for the inherent differences in the implementation of vaccination programs at the municipal level. Model parameters were presented as odds ratios (ORs) with 95% confidence intervals (CIs). An assessment of the adherence to model assumptions and the overall model fit was conducted. All analyses were performed in STATA/SE version 15.1 (StataCorp LLC, College Station, TX, USA) [17].

This study was given ethical clearance by the UP Manila Research Ethics Board (UPMREB).

## 3. Results

A total of 775 caregivers participated in the survey. The respondents, aged 18 to 73 with a median age of 32 years, 752 (97.0%) were females, 689 (90.1%) were married or in a live-in relationship, 649 (83.7%) were Catholics, 461 (59.5%) were from higher income municipalities, 668 (82.5%) had elementary or high school education, and 455 (58.7%) were unemployed (Table 1).

Among the respondents, 560 (72.3%) completed the primary series with or without booster doses. There were 586 (75.6%) respondents who received at least one dose of the COVID-19 vaccine, while 189 (24.4%) were unvaccinated (Table 2).

Among the 586 respondents who received at least one dose of COVID-19 vaccine, the protection of self, family, and community was the leading reason for vaccination (93%), followed by the influence of government-mandated regulations and restrictions (40.6%) (Table 3). The most common reason (81.4%) for non-vaccination was distrust in vaccine safety including fear of side effects (Table 3). Among the 3.3% of respondents with an incomplete primary vaccine series, fear of side effects was the main reason (23.1%) stated for not pursuing full vaccination (Table 3).

Among caregivers with complete primary series, DOH announcements (72.1%), mass media outlets (56.3%), and local government units (LGUs)/health centers and healthcare workers (HCWs) (51.4%) were identified as the main sources of information on COVID-19 vaccination. Conversely, those unvaccinated or who received incomplete primary series obtained information from mass media outlets (64.7%) and the DOH (46.5%), but with social media (38.1%) among their top three information sources (Table 4).

The majority of respondents who completed the primary series considered COVID-19 vaccines to be important (84.5%) and safe (80%). A significant proportion were also satisfied with COVID-19 vaccine efforts in their area (80%) (Figure 1). A third of the respondents who were unvaccinated or received an incomplete primary series thought that COVID-19 vaccines can cause negative side effects (34.4%). They were also of the opinion that COVID-19 is only a mild disease (31.7%) and is not real (24.7%) (Figure 1).

The logistic regression analyses showed that the odds of completing a primary series of COVID-19 vaccination increases with older age [30–45 years (aOR = 2.23; 95% CI 1.49–3.35) and 46–59 years (aOR = 2.84; 95% CI 1.36–5.95)], higher education level [secondary (aOR = 2.25; 95% CI 1.47–3.43) and tertiary (aOR = 4.93; 95% CI 2.37–19.27)], and being employed (aOR = 1.99; 95% CI 1.24–3.19). Some perceptions about COVID-19 vaccination related to vaccine safety, effectiveness, COVID-19 vaccination efforts, and policies were also associated with the said outcome. Perceptions which served as enablers of COVID-19 vaccination included agreement with the following statements: “COVID-19 vaccines are safe.” (aOR = 1.92, 95% CI 1.16–3.18) and “I am satisfied with COVID-19 efforts in my area.” (aOR = 2.39, 95% CI 1.14–4.06). Disagreement with the statement “Vaccines cannot protect me from severe COVID-19 disease.” also served as enabler (aOR = 2.23, 95% CI 1.38–3.63) of having a primary series of COVID-19 vaccination. On the other hand, a significant barrier to COVID-19 vaccination was related to practical issues. Those who disagreed with the statement “Vaccination should be required in schools and workplaces.” had significantly lower odds (aOR = 0.31, 95% CI 0.18–0.55) of completing the primary series of COVID-19 vaccination compared to those who were neutral to the statement (Table 5).

## 4. Discussion

This study explored the enablers and barriers of COVID-19 vaccine uptake among primary caregivers. The most important factors influencing the uptake of vaccines were identified as age, educational attainment, and employment status, as well as vaccine confidence, satisfaction with COVID-19 efforts, and agreement with vaccination-related policies. The top reasons for completing the COVID-19 primary series were the protection of self, family, and community; influence of government-mandated regulations; and recommendation of family, friends, and neighbors. Conversely, commonly cited reasons for non-vaccination included distrust in vaccine safety or a fear of side effects, a lack of time, and being pregnant.

The study sites were considered to have lower vaccine coverage since they did not meet the national target. However, 75.6% of the respondents received at least one dose of COVID-19 vaccine. This may be attributed to the difference between the period wherein the baseline data were obtained (August 2022) and the period of data collection (July–October 2023). The most common reasons for vaccination were the protection of self, family, and community and the influence of government-mandated regulations. Similar to other studies [18,19], these results suggest that the respondents value the protection provided by the vaccines. The findings also highlighted the positive effect of governance across various administrative levels. Governmental interventions play a crucial role in improving COVID-19 vaccination rates [20,21], but such measures need to be implemented prudently where ethical norms are safeguarded [21]. Other reasons for vaccination were recommendations from family, friends, and neighbors. Individuals were motivated to get vaccinated by observing others including relatives, neighbors, and prominent people, such as community leaders, receive their COVID-19 vaccines. This highlights the role of an individual’s social network in shaping actions and behaviors [22]. The provision of incentives was also a motivating factor for some adults to submit themselves to vaccination. One of the strategies employed in some low- and middle-income countries to increase vaccine acceptance was to provide financial or non-financial rewards [11].

In contrast, a minority (24.4%) were unvaccinated against COVID-19. The primary reason for their decision was distrust in vaccine safety or a fear of side effects. This hesitation could be attributed to the novelty of the vaccine, potentially compounded by previous controversies such as the Dengvaxia issue [23]. Supporting evidence from other countries also revealed that concerns about the adverse effects of COVID-19 vaccines were among the top reasons for vaccine hesitancy [24,25,26]. While a lack of time was reported by only a small fraction, it notably emerged as another reason for non-vaccination among respondents, aligning with findings from another study [27]. Furthermore, being pregnant was also a reason for non-vaccination. This can be attributed to the characteristics of the respondents in this study, being predominantly females. In a similar study, pregnancy was viewed as a barrier for COVID-19 vaccination since respondents were uncertain about the effects of the vaccine on their fetuses [22].

In this study, the majority of respondents who completed the COVID-19 primary series had a high perceived need of vaccination (84.5%) and believed that COVID-19 vaccines are safe (80%). Most of them also showed satisfaction with COVID-19 vaccination efforts in their area (80%). The common sources of information for these respondents included the DOH and LGUs/health centers and HCWs, which is similar to findings from other studies [28,29]. This shows that there is a high level of confidence in health authorities, including HCWs, and government agencies. These information sources played a role in shaping their perceptions on COVID-19 infection and vaccination [30].

On the other hand, a significant proportion of respondents who were unvaccinated or received an incomplete COVID-19 primary series had low vaccine confidence (34.4%) and low perceived disease risk (31.7%). The rapid development of COVID-19 vaccines raised questions about their safety and long-term side effects and might have contributed to the low vaccine confidence. Concerns on side effects like blood clots and fertility issues experienced by family members or friends further added to vaccine hesitancy [31,32]. Low perceived disease risk can stem from individuals believing they are healthy enough to easily combat the virus and no longer view COVID-19 as a significant threat [31]. The perceptions of unvaccinated respondents may have been shaped by social media, being one of the primary sources of information, where misinformation and disinformation is being proliferated.

In the logistic regression analysis, sociodemographic characteristics positively associated with COVID-19 vaccine uptake were older age, specifically the 46–59 years of age group, higher educational attainment, and being employed. Older respondents were more inclined to be part of the employed population subjected to mandatory vaccination protocols during the pandemic. Employment was consistently an enabler of vaccine uptake as vaccination was mandatory for employees during the pandemic, as seen in other studies [8,9,33]. While older age has been identified as a contributing factor to higher vaccine uptake in various studies [10], some studies found that it was a barrier [7,8], possibly due to poor health (e.g., the presence of more chronic conditions) and concerns about vaccine side effects [34]. Higher educational attainment may have contributed to respondents’ better understanding of the benefits of novel COVID-19 vaccines, potentially enabling vaccine uptake.

Other enablers found to be significant in this study were vaccine confidence, satisfaction with the implementation of the COVID-19 vaccination program, and agreement with making vaccination a requirement. These findings were consistent with other studies [35,36,37], highlighting the importance of public trust in vaccines, policies, and governance.

The main limitation of this study is the non-random selection of study sites, which may have affected the generalizability of our findings. Another limitation is the representativeness of the study, as certain regions were excluded, despite having low vaccine uptake, due to logistical constraints and safety concerns in conducting the study. Potential bias may have occurred due to the gender distribution of the respondents being predominantly female. This is due to the inclusion criteria of being a primary caregiver to children aged 0–11. Social desirability bias may have occurred as respondents completed the survey in the presence of the research team and their peers, thereby, providing expected answers. The vaccination status of the respondents was self-reported as medical records or proofs of vaccination were not presented during the conduct of the survey. Geographic differences between the study sites were not explored further as only the municipal class was used as a variable. Other geographic factors (e.g., urban/rural, population density, and the remoteness of the location) were not obtained. Vaccine preferences were also not elaborated as the limited supply of vaccines during the roll-out left people with no choice but to accept what was available in the vaccination center. Lastly, the directionality of the relationship between the beliefs regarding vaccination and vaccine uptake could not confirmed due to the cross-sectional study design.

Despite these limitations, this study identified the multifaceted factors influencing COVID-19 vaccination uptake among adults in the Philippines, underscoring the pivotal role of governance, social influence, and individual perceptions. It highlights the need for targeted interventions addressing vaccine hesitancy among specific demographic groups while emphasizing the importance of fostering public trust, satisfaction with vaccination efforts, and alignment with vaccination policies to enhance overall vaccine confidence and coverage. This constitutes important evidence for enhancing vaccination policy and roll-out in preparation for future outbreaks or pandemics.

## 5. Conclusions

This study highlights the enablers and barriers of COVID-19 vaccine uptake among primary caregivers in regions in the Philippines with low vaccine uptake. Perceptions on the benefits and the safety of vaccines play a large role in the decision of the respondents to vaccinate. Satisfaction with the implementation of vaccination programs was also highlighted as an enabling factor. We recommend that health promotion and education must be timely and must focus on key messaging that highlights benefits and addresses misconceptions regarding safety concerns including the possible side effects of vaccination. The findings of this study on COVID-19 vaccine uptake may provide evidence for strategies for enhancing vaccine confidence during the introduction of novel vaccines and the threat of future pandemics.

## Figures and Tables

**Figure 1 vaccines-13-00719-f001:**
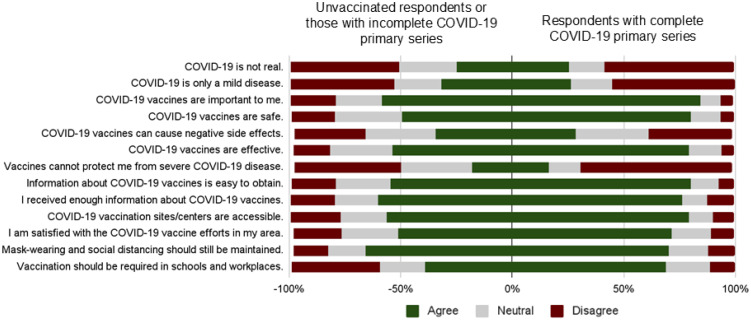
Perceptions on COVID-19 infection and vaccination among respondents with or without complete COVID-19 primary vaccination.

**Table 1 vaccines-13-00719-t001:** Sociodemographic characteristics among all survey respondents (n = 775) *.

	n	%
Age		
18–29	290	37.5
30–45	382	38.4
46–59	86	11.1
≥60	16	2.1
Sex		
Female	752	97
Male	20	2.6
Civil status		
Single	25	3.2
Married	371	47.9
Live-in	327	42.2
Separated	14	1.8
Widowed	25	3.2
Unspecified	13	1.7
Municipal Class		
First Class Municipality	263	33.9
Second Class Municipality	198	25.6
Third Class Municipality	260	33.6
Fifth Class Municipality	54	6.9
Religion		
Catholic	649	83.7
Other Christian denominations	110	14.2
Islam	9	1.2
Others	6	0.8
Educational attainment		
Elementary Level	210	27.1
High School Level	458	59.1
College Level	107	13.8
Employment status		
Unemployed	455	58.7
Student	6	0.8
Pensioner	5	0.6
Self-employed	75	9.7
Employed	233	30.1

* Some percentages do not add up to 100% due to missing data.

**Table 2 vaccines-13-00719-t002:** COVID-19 vaccination status among all survey respondents (n = 775).

	n	%
Unvaccinated	189	24.4
Incomplete primary COVID-19 vaccine series	26	3.3
Complete primary series	443	57.2
Complete primary series with one booster dose	79	10.2
Complete primary series with two booster doses	38	4.9

**Table 3 vaccines-13-00719-t003:** Reasons for complete, incomplete, and non-vaccination for COVID-19.

Reasons for Vaccination Among Those Respondents Who Received at Least One Dose (n = 586) *	n	%
Protection of self, family, and community	545	93
Influence of government-mandated regulations ****	238	40.6
Recommendation of family, friends, neighbors, etc.	35	6
Provision of incentives and rewards	15	2.6
Requirement for giving birth	2	0.3
Unspecified	4	0.7
**Reasons for non-vaccination among unvaccinated respondents (n = 189) ***	**n**	**%**
Distrust in vaccine safety/fear of side effects	154	81.4
Lack of time	15	7.9
Being pregnant	14	7.4
Doubt in vaccine effectiveness	10	5.3
Perception of COVID-19 vaccines being in experimental stage	8	4.2
Influence of religion/culture	7	3.7
Dislike of vaccine brand offered in the area	6	3.2
Perception that COVID-19 is not severe	4	2.1
Inaccessibility of vaccination sites	2	1.1
Others ***	6	3.2
**Reasons for incomplete vaccination among respondents with incomplete primary vaccine series (n = 26) ***	**n**	**%**
Fear of side effects	6	23.1
Low perceived need of vaccination	3	1.2
Inaccessibility of vaccination sites	3	1.2
Doubt in vaccine effectiveness	3	1.2
Difficulty in arranging schedules/long queues	2	0.8
Lack of time	2	0.8
Doubt in the quality of vaccine	1	0.0

* Multiple answers allowed. ** Includes requirement for school, work, leisure, and travel. *** Respondent being sick, negative information circulating in the community, and fear of injection.

**Table 4 vaccines-13-00719-t004:** Sources on information on COVID-19 vaccination *.

	Unvaccinated or Incomplete Primary Series (n = 215)	Complete Primary Series with or Without Boosters (n = 560)	Total (n = 775)
DOH announcements	100 (46.5%)	404 (72.1%)	504 (65%)
Mass media outlets	139 (64.7%)	315 (56.3%)	454 (58.6%)
LGU/health center and healthcare workers	69 (32.1%)	288 (51.4%)	357 (46.1%)
Social media	82 (38.1%)	198 (35.4%)	280 (36.1%)
Online articles/pages of legitimate news outlets	20 (9.3%)	67 (12%)	87 (11.2%)
International agencies (e.g., WHO and CDC)	13 (6.1%)	54 (9.6%)	67 (8.7%)
Family and friends	21 (9.8%)	41 (7.3%)	63 (8.1%)
Own reading/research	11 (5.1%)	33 (5.9%)	44 (5.7%)
School	10 (4.7%)	22 (3.4%)	32 (4.1%)
Church and religious leaders	10 (4.7%)	17 (3%)	27 (3.5%)
Workplace	2 (0.9%)	11 (2%)	13 (1.7%)
Celebrities and influencers	2 (0.9%)	8 (1.4%)	10 (1.3%)
Tribal chief/leader	3 (1.4%)	5 (0.9%)	1 (0.1%)
Unspecified	3 (1.4%)	5 (0.9%)	9 (1.2%)
No information	1 (0.5%)	1 (0.2%)	2 (0.3%)

* Multiple answers allowed.

**Table 5 vaccines-13-00719-t005:** Factors associated with COVID-19 vaccine uptake (n = 775).

Sociodemographic Characteristics and Perceptions on COVID-19 Infection and COVID-19 Vaccination	n (%) ^†^	Crude OR (95% CI)	Adjusted OR Final Model ^††^ (95% CI)
Age			
18–29	290 (37.5)	1.0	1.0
30–45	382 (38.4)	2.37 (1.69–3.32) ***	2.23 (1.49–3.35) ***
46–59	86 (11.1)	4.05 (2.11–7.79) ***	2.84 (1.36–5.95) **
≥60	16 (2.1)	1.97 (0.62–6.26)	1.41 (0.33–6.06)
Educational attainment			
Elementary level	210 (27.1)	1.0	1.0
High school level	458 (59.1)	2.27 (1.61–3.21) ***	2.25 (1.47–3.43) ***
College level	107 (13.8)	4.89 (2.62–9.13) ***	4.93 (2.37–19.27) **
Employment status			
Unemployed	455 (58.7)	1.0	1.0
Student	6 (0.8)	2.51 (0.29–21.66)	1.39 (0.12–15.81)
Pensioner	5 (0.6)	0.75 (0.12–4.55)	0.21 (0.03–1.80)
Self-employed	75 (9.7)	1.48 (0.85–2.58)	1.27 (0.64–2.46)
Employed	233 (30.1)	2.35 (1.59–3.47) ***	1.99 (1.24–3.19) **
Statements on COVID-19 infection and COVID-19 vaccination			
COVID-19 vaccines are safe.			
Neutral	139 (17.9)	1.0	1.0
Disagree	75 (9.7)	0.73 (0.41–1.28)	0.93 (0.47–1.85)
Agree	554 (71.5)	3.71 (2.50–5.51) ***	1.92 (1.16–3.18) *
Vaccines cannot protect me from severe COVID-19 disease.			
Neutral	146 (18.8)	1.0	1.0
Disagree	485 (62.7)	3.23 (2.19–4.78) ***	2.23 (1.38–3.63) **
Agree	132 (17.0)	2.08 (1.27–3.41) **	1.58 (0.86–2.88)
I am satisfied with COVID-19 vaccination efforts in my area.			
Neutral	122 (15.7)	1.0	1.0
Disagree	82 (10.6)	0.73 (0.42–1.28)	1.34 (0.67–2.68)
Agree	565 (72.9)	2.94 (1.95–4.40) ***	2.39 (1.14–4.06) **
Vaccination should be required in schools and workplaces.			
Neutral	154 (19.9)	1.0	1.0
Disagree	146 (18.8)	0.28 (0.17–0.45) ***	0.31 (0.18–0.55) ***
Agree	470 (60.7)	1.78 (1.17–2.72) *	1.10 (0.65–1.86)

^†^ Percentages do not add up to 100% due to incomplete or invalid answers. ^††^ Adjusted model accounts for the effect of municipality. Level of significance: * *p* < 0.05, ** *p* < 0.01, and *** *p* < 0.001.

## Data Availability

The datasets used and/or analyzed during the current study are available from the corresponding author on reasonable request.

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
