# Peer review of "Enablers and Barriers of COVID-19 Vaccination in the Philippines"

_vaccines, 2025, doi:10.3390/vaccines13070719_

Round 1

Reviewer 1 Report

Comments and Suggestions for Authors

Thank you for the opportunity to review the manuscript ID: vaccines-3704831. This study aimed to identify factors that affect COVID-19 vaccine uptake among primary caregivers in regions with low vaccine uptake in the Philippines.

Comments

Section Introduction:

Describe the circumstances in which the presented study was carried out (with citation of appropriate references):
Describe the epidemiological situation of COVID-19 in the Philippines, including incidence and mortality, with age distribution.
Describe when vaccination against COVID-19 began in the Philippines, including indications for vaccination, whether it is free or paid. Also, list the coverage of vaccination against COVID-19 in the Philippines, with the distribution of those vaccinated by age.

Methods section:

Line 73: Define `lower COVID-19 vaccine coverage`.
Lines 83-88: State whether there were any criteria for excluding participants for this study.
Lines 98-100: Indicate whether the survey was anonymous.
Line 100: State whether there were any missing answers, that is, incompletely filled questionnaires, and how they were addressed in this analysis.

Results section

Table 1: Match the numbers and percentages according to the total number of participants in this study (that is `... a total of 775 caregivers participated in the survey.`) in the appropriate way.

Table 5: Check and mark the significance for the `Agree` category for the variable `Vaccination should be required in schools and workplaces.`.

Discussion section

Lines 183-184: Explain the finding that this study was carried out in places with `lower COVID-19 vaccine coverage`, while the results of this work identified the coverage of 75.6%, I quote this sentence `In this study, the majority of the respondents (75.6%) received at least one dose of COVID-19 vaccine`.

Lines 245-250: In addition to the aforementioned, list and discuss numerous other limitations of this study (such as study design, lack of access to medical records to check vaccination status, etc.).

Reviewer 2 Report

Comments and Suggestions for Authors

This manuscript explores the enablers and barriers of COVID-19 vaccine uptake among primary caregivers in selected low-coverage regions in the Philippines. The study is grounded in a clear methodology, uses an appropriate tool, and offers actionable recommendations for public health strategy and outreach. The sample size is robust, and the findings contribute to vaccine equity literature in low- and middle-income country (LMIC) settings.

A few methodological and interpretive issues should be addressed to strengthen the manuscript's clarity, generalisability, and impact.

Comments.
1. Table 2, Dose-based sorting:
The ordering of vaccination categories in Table 2 is alphabetical (A to Z), which can be confusing.

Consider reordering categories by vaccine dose hierarchy for clarity (0 to 4): Unvaccinated → Incomplete primary series → Complete primary series → Complete primary + one booster → Complete primary + two boosters.

2. Geographic distribution and access barriers:
The Philippines is an archipelagic country with over 2,000 inhabited islands. Many populations live in geographically isolated and disadvantaged areas (GIDAs), which pose challenges to cold-chain logistics, vaccine transport, and workforce deployment.

 Consider classifying baseline characteristics by urban, suburban, and remote locations or by major vs. remote/isolated islands. If geographic classification data are available, comparisons across these subgroups (e.g., as supplementary tables) may reveal critical insights into disparities in access and uptake.

3. Single timepoint (Cross-sectional) design:
The study uses a cross-sectional design with data collection during July–October 2023. This makes it difficult to assess directionality (i.e., whether beliefs influenced vaccine uptake or vice versa).

Suggest discussing this limitation more fully in the Discussion. Additionally, note that this period corresponds to the Omicron variant with various subvariant waves, which may have influenced risk perception or urgency around vaccination.

4. Self-reported data and potential biases:
All data in this manuscript are based on self-reports, which raises concerns about recall bias (e.g., about specific beliefs before vaccination) and social desirability bias (e.g., over-reporting vaccination if it was perceived as socially desirable), especially when reporting vaccination status or health-seeking behaviours.

Suggest acknowledging this more prominently in the Limitations and clarify whether any validation (e.g., health records or vaccine cards) was feasible.

5. Definition of "primary caregiver":
The manuscript refers to “primary caregivers” but does not define how this was determined.

Clarify whether participants self-identified as primary caregivers or whether an operational definition (e.g., decision-maker for the child’s health) was applied.

6. COVID-19 vaccine types:
The Philippines deployed multiple vaccine types during its rollout, such as CoronaVac (Inactivated), ChAdOx1-S (viral vector), BNT162b2 (mRNA).

If data on vaccine type are available, please consider reporting or discussing the influence of brand perception on uptake. If not available, acknowledge this as a limitation, especially given prior issues with vaccine trust in the country (e.g., Dengvaxia controversy).

Errors.
1. Tables 1 & 5, age category format:
The category “>60” is unclear. If this includes participants aged 60 and above, revise to “≥60” for clarity and consistency.

Reviewer 3 Report

Comments and Suggestions for Authors

Overall Comment

The authors conducted this study to evaluate the enablers and barriers influencing COVID-19 vaccination among primary caregivers in Philippines. While the manuscript identifies relevant factors, such as age, education levels, employment status, vaccine confidence, that are associated with vaccine uptake, these factors have already been widely reported in existing literatures, including in studies cited by the authors themselves. As such, the study does not considerably advance the current understanding of the dynamics related to COVID-19 vaccine acceptance.

Moreover, compared to other similar studies, this manuscript does not appear to include a sufficiently large respondent pool or survey points that would enable to generate more robust or impactful data. Considering the authors’ own stating of limitation regarding non-random selection of study sites, the limited sample size further restricts the potential for meaningful novel findings.

In conclusion, although the study is methodologically sound in its descriptive design, the limited originality and lack of new insights substantially reduce its contribution to the field.

Round 2

Reviewer 1 Report

Comments and Suggestions for Authors

Thank you for the opportunity to re-review the manuscript ID: vaccines-3704831.
The authors correctly addressed all my comments and corrected this paper in an appropriate way. I thank the authors.     

Author Response

Thank you for all your valuable comments and suggestions which contributed to the further improvement of our manuscript. 

Reviewer 2 Report

Comments and Suggestions for Authors

Thank you for thoroughly addressing the concerns I raised in your previous submission. I found only one point that you may have forgotten to revise.

Errors.
Tables 1 & 5, age category format: The category “> 60” is unclear. If this includes participants aged 60 and above, revise to ≤“60” for clarity and consistency

Author Response

Thank you for all your valuable comments and suggestions which contributed to the further improvement of our manuscript. The category “>60” was changed to “≥60” in Tables 1 and 5.

Reviewer 3 Report

Comments and Suggestions for Authors

I agree with the authors’ response emphasizing the importance of context-specific evidence and the unique relevance of this study in the cohort setting, particularly given the historical and social factors influencing vaccine confidence.

As an additional suggestion, in the discussion of study limitations, the authors might consider briefly elaborating on the gender distribution of the cohort. While the cohort being predominantly female is understandable given the circumstances, explicitly noting this potential gender bias as a limitation could help readers develop a more comprehensive understanding of the study findings. This is not a mandatory revision but rather a recommendation for possible enhancement.

Author Response

Thank you for your comments and suggestions which contributed to the further improvement of our manuscript. Discussion on the potential gender bias is now included in the limitations. This can be found in Lines 270-273: 

"Potential bias may have occurred due to the gender distribution of the respondents being predominantly female. This is due to the inclusion criteria of being a primary caregiver to children aged 0-11."